# $\alpha$-FORMER:LOCAL-FEATURE-AWARE(L-FA) TRANSFORMER

## ABSTRACT

Despite the success of current segmentation models powered by the transformer, the camouflaged instance segmentation (CIS) task remains a challenge due to the similarity between the target and the background. To address this issue, we propose a novel approach called the local-feature-aware transformer ($\alpha$-Former) that incorporates traditional computer vision descriptors to extract critical edge features in camouflaged instances. Specifically, we introduce an adapter to merge local features into the transformer framework. Using the proposed transformer-based encoder-decoder architecture, our $\alpha$-Former surpasses state-of-the-art performance on the COD10K and NC4K datasets. Additionally, we introduce an edge-aware feature fusion module to improve boundary results in the segmentation model.

## 1 INTRODUCTION

Camouflaged instance segmentation (CIS) is beneficial for applications in computer vision, like medical image segmentation, agriculture, etc. (Fan et al. (2020)). However, this task is challenging compared to traditional object detection and segmentation since camouflaged objects can effectively blend in with the background, making it difficult for models to detect and annotate them accurately. Recently, transformer reached outstanding performances in different applications like classification (Chen et al. (2021), detection (Carion et al. (2020)), segmentation (Strudel et al. (2021)), etc. However, transformer models usually need a large-scale dataset for training. Thanks to the large-scale camouflaged datasets and benchmarks like COD10K (Fan et al. (2020), CAMO ( Le et al. (2019)), CAMO++ (Le et al. (2021)), NC4K (Lv et al. (2021)), it is possible for the researchers to implement the transformer on CIS. As a result, the transformer achieves state-of-the-art performance in this field (Pei et al. (2022)).

Despite their effectiveness, current transformer models have limitations in dealing with camouflaged instance segmentation. As shown in Fig 1, these models tend to predict multiple objects for a single target when the edge is unclear. This is because the models primarily focus on finding the target object and ignore the importance of accurately identifying the boundary of the target object. To improve camouflaged instance segmentation performance, models need to better understand the object's location and enhance the features around the instance's boundary.

In order to improve the boundary features of our model, we have integrated traditional descriptors like LBP( Ojala et al. (1994)) into the transformer framework. LBP is especially sensitive to edges, which is advantageous in the context of CIS because of the high similarity between objects and the background. As depicted in Figure 1, LBP can accurately demarcate the boundary of the target object, even when the color and texture of the object are similar to that of the background. This allows the model to achieve superior results, as shown in Figure 1. By combining LBP with the transformer, we have developed an effective framework for identifying target objects and creating precise boundaries. We call this framework the local-feature-aware transformer, or $\alpha$-Former (pronounced "alpha-former"). Inspired by LBP, we have created a learnable module known as the binary filter (BF), which can compare pixel values within a field and generate a local feature. The binary filter consists of a learnable module and a fixed-weight convolution layer called BCNN which can extract features similar to the LBP.

The fixed convolution layer is able to generate local features by comparing different pairs of pixels, while the learnable module can collect and consolidate this comparison information. To effectively

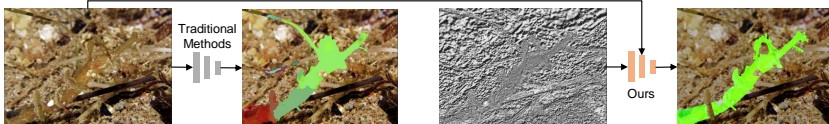

Figure 1: The $\alpha$-Former was motivated by the need to improve the performance of the camouflaged instance segmentation model. The model generates a local feature that provides precise boundary information about the target object. The first image is the input image, the second image is the prediction result without the local feature, the third is the generated local feature, and the fourth is the prediction result with the local feature. Incorporating the local feature into the model results in a more accurate segmentation of the target object.

integrate the features extracted by the binary filter, we have developed a learnable module known as the feature aggregation adapter (FAA). The FAA can provide the local features to the backbone of the model without interfering with its performance, even if there are differences in the input distribution. Moreover, our FAA module is highly parameter-efficient and easy to train. Additionally, we have designed an edge-aware module that can accurately predict boundaries for CIS. This module includes a multi-level convolution layer that offers a wide receptive field and a fixed-weight convolution layer that extracts local features. To utilize the ground truth edge as supervision, we employ a $1 \times 1$ convolution layer to generate edge predictions. These edge predictions are then incorporated into the final prediction head to improve the model's overall performance.

Our model combines the binary filter (BF), feature aggregation adapter (FAA), and edge-aware fusion module to achieve superior performance on two popular datasets, COD10K and NC4K. Specifically, our model outperforms the current state-of-the-art by approximately 2 average precision (AP) points. Additionally, we conduct ablation studies to demonstrate the effectiveness of our BF, FAA, and edge-aware fusion module. Also, we provide lots of qualitative results in our experiments.

To summarize, our contributions are:

- We first notice that some traditional descriptors are sensitive to the boundary of camouflaged objects. Inspired by the finding, we proposed a learnable module to extract similar features as the traditional descriptor.
- We proposed $\alpha$-Former, which firstly provides local binary information to the camouflage instance segmentation model. Also, we provide edge supervision to our model to improve the final mask boundary.
- We achieve state-of-art camouflaged instance segmentation performance on two different datasets. Experiments and ablation study shows the effectiveness of our proposed modules and architecture.

## 2 RELATE WORK

**Camouflaged Object Detection.** Camouflaged object detection aims to find the hidden object in the image and is more challenging than traditional object detection. Earlier works mainly focus on some level features like color (Huerta et al. (2007)), and texture (Song & Geng (2010)). With the development of deep learning, more and more works have started to use neural networks to solve the problem. These methods (Zhu et al. (2021); Mei et al. (2021)) mostly use a CNN backbone to extract high-level features and try to locate and predict the camouflaged objects. MGL (Zhai et al. (2021)) first use a mutual graph to detect and predict the final results. UGTR (Yang et al. (2021)) tried to mimic the human process, adding an uncertain prediction for camouflaged object detection. OSFormer (Pei et al. (2022)) uses a one-stage architecture and transformer to get the final results. PFNet (Mei et al. (2021)) firstly adds a positioning and focus module to mimic the human detection process, which tries to find the target object.

**Integrating traditional descriptors to Help CNN.** There is a long history of using traditional descriptors to help improve the performance of CNN. Earlier works use different descriptors to help CNN. For example, some works (Karanwal & Diwakar (2021a;b)) use LBP (Ojala et al. (1994)) to help improve face performance recognition. People also use HOG (Dalal & Triggs (2005)) to help

Figure 2: Examples of our BCNN layer. The left is a sample of $3 \times 3$ BCNN layer, the center is the input image, and the right is the output of the BCNN layer. Our results show that BCNN can provide a precise boundary for a given image.

them improve the performance of human detection (Surasak et al. (2018)) and action recognition (Patel et al. (2020)). Recently, researchers tried to combine SIFT (Lowe (1999)) and convolution networks (Gupta et al. (2019); Hossein-Nejad et al. (2021); Kovač & Marák (2022)) to extract better features and implement the features in different applications. Considering so many works integrating traditional descriptors with deep learning architecture and get performance improvement and the lack of effort to apply the traditional descriptor to camouflaged object detection, we try to use a descriptor inspired by traditional descriptors to improve the performance of camouflaged object detection.

**Binary Filter.** The traditional descriptor inspires the idea of using a binary filter for convolution. Many works already use their binary filter to get good performance in many datasets. For example, BinaryConnect (Courbariaux et al. (2015)) tried to train a neural network with only binary weights during propagation. In this article, they approximate the real value in neural networks with binary values. Based on BinaryConnect, researchers proposed BinaryNet (Courbariaux et al. (2016)), where both the weights and activations are constrained to $+1$ or $-1$. LBCNN (Juefei-Xu et al. (2017)) uses a fixed-weight binary convolution to replace the original convolution and get good performance in the classification tasks. These works show the feasibility of using binary filters to extract features and train neural networks.

# 3 BINARY FILTER

## 3.1 WHY USE BINARY FILTER

We have observed that traditional camouflage segmentation models struggle to determine the boundary of objects in ambiguous cases accurately. For example, when presented with an image of a pipefish, as shown in Fig. 1, a standard model may predict multiple objects instead of correctly identifying a single target object. However, using a traditional descriptor like LBP, as shown in Fig. 1, enables the model to locate the object's boundary. The resulting feature representation is also continuous, encouraging the model to predict the target object as a single entity instead of multiple objects. Unfortunately, LBP is not a learnable descriptor, meaning that it cannot adapt to new data.

To address this issue, we sought to design an architecture that can detect local binary features similar to those captured by traditional descriptors but is also learnable. The LBP descriptor compares the center pixel value with the surrounding pixel values, so we were inspired to create a binary filter using a fixed binary weight convolution (BCNN) to simulate this process.

## 3.2 ARCHITECTURE OF BINARY FILTER

In this section, we describe the architecture of our proposed binary filter, which allows for comparison operations that are difficult to perform with traditional convolution layers. As illustrated in Fig. 2, we can simulate the comparison operation by designing a convolution kernel where the center value is -1, the left value is 1, and all other values are 0. After applying this convolution operation, we compare the output with 0. If the output is greater than 0, we know that the left pixel value is greater than the center pixel value; otherwise, the left pixel value is less than the center pixel value. Our designed binary convolution layer with fixed binary weight convolution (BCNN) can extract the precise boundary for the target object, as demonstrated in Fig. 2. To increase the robustness of BCNN, we use multiple binary convolution kernels for each BCNN layer, and for each kernel, we randomly select a value from $-1, 0, 1$. However, the BCNN is not trainable, and to make the binary filter trainable, we add a $1 \times 1$ convolution layer after each BCNN layer to gather information, which

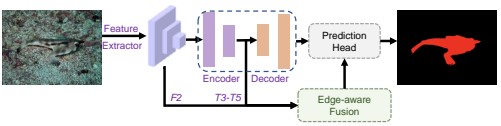

Figure 3: $\alpha$-Former comprises a feature extractor, an encoder-decoder, an edge-aware fusion module, and a prediction head. The input of the $\alpha$-Former is a single RGB image, and $\alpha$-Former can segment the camouflaged object in the input image.

is trainable. This trainable $1 \times 1$ convolution layer is very light and easy to train compared to the traditional CNN architecture.

## 4 METHODS

**Architecture** Our proposed $\alpha$-Former has five crucial modules. (1) A feature extractor with a binary filter to extract similar features as the LBP( Ojala et al. (1994)), an adapter to transfer the input domain, and a backbone to extract object features. (2) A transformer encoder that uses global and local features to generate object embedding. (3) An edge-aware feature fusion module to generate precise boundaries. (4) A transformer decoder to extract the information from the embedding (5) Mask predict head to predict final instance mask. The whole architecture is shown in Fig.3

### 4.1 FEATURE EXTRACTOR

Our feature extractor consists of three parts: a learnable local binary filter (BF), a feature aggregation adapter (FAA), and a pre-trained CNN backbone. These components are shown in Fig.4.

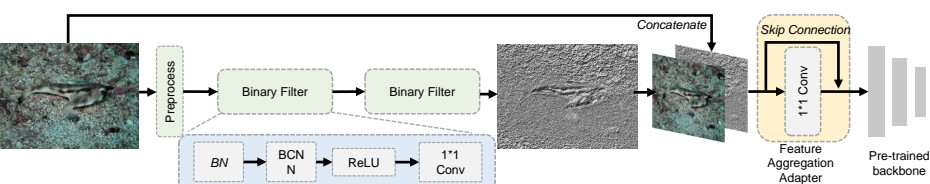

Figure 4: Our feature extractor contains a preprocessing module, several binary filters, a feature aggregation adapter, and a pre-trained backbone. The binary filter can extract local features of the input image. After getting local features, we concatenate the original image and local features and use our feature aggregation adapter to transfer the new image domain to the input image domain. After the feature aggregation adapter, we use a pre-trained backbone to extract high-level and low-level features.

### 4.1.1 BINARY FILTER (BF)

The purpose of the binary filter is to extract local features. Here, given an input image $I \in \mathbb{R}^{H \times W \times 3}$, we firstly use a convolution layer to preprocess the image. After the preprocessing, we use a pre-defined binary filter to extract local binary features. The detail of the binary filter is already discussed in Sec.3. In every experiment, we use multiple binary filters to extract the local binary information. After the BF module, we can get a feature $F \in \mathbb{R}^{H \times W \times C}$ where C is the channel number of the final $1 \times 1$ convolution. Then we concatenate the original image $I$ and the feature $F$.

### 4.1.2 FEATURE AGGREGATION ADAPTER (FAA)

After the BF module, the channel numbers of concatenate images are different from the backbone training images, which makes it not practical to use the pre-trained backbone directly. In order to use the pre-trained backbone, we need a method to transfer the concatenated image to the same domain as the original images. Here, we introduce a feature aggregation adapter to transfer the new image domain to the original image domain. The architecture of the adapter is a $1 \times 1$ convolution and a skip connection, which can be seen in Fig.4. The output shape of the adapter is $H \times W \times 3$, which

is the same as the original images. The purpose of adding a skip connection is that, at the beginning of the training, it is challenging to initialize the parameter of the $1 \times 1$ convolution to guarantee the domain of the output is the same as the domain of the original image. In order not to influence the performance of the backbone at the beginning of the training, we can set very tiny initial values of the $1 \times 1$ convolution layer. Furthermore, we can directly add the first three channels, the original images, to the output for the skip connection. This operation can ensure the input of the backbone is almost the same as the original image at the beginning of the training. During training, the model can gradually learn to use the local binary features.

### 4.1.3 CNN BACKBONE

We use a pre-trained backbone in our experiments. In order to provide low-level features and high-level features to the prediction module, we use multi-scale features from the backbone. We will use most of our experiments' last four layers' features. We will use $F_2 - F_5$ to represent different layer features in the following parts. Because the backbone's input contains more local features than the original image, the extracted features of the backbone contain extra information compared to directly inputting the original images to the backbone.

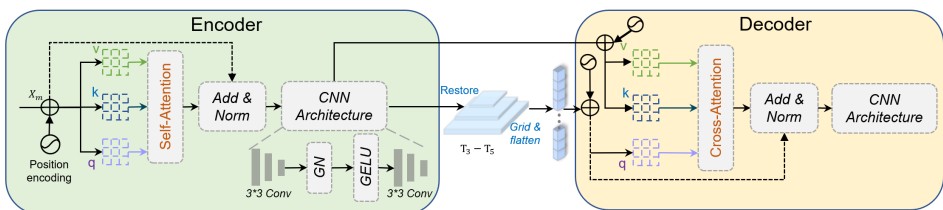

Figure 5: Our encoder contains a position encoding module, a self-attention module, and a CNN architecture. The encoder's input is the extracted third to fifth layer's backbone features. After getting the input feature, we first add a position embedding to the features and then use a self-attention module to get its local features. After getting the local feature, we use an add & norm operation followed by a CNN architecture to get the final output of the encoder. Then we restore and grid the output of the encoder to a location-aware query and input the query to the decoder. In the decoder, we use a cross-attention module to extract information. After the cross-attention, we use the same CNN architecture as the encoder.

### 4.2 ENCODER

In order to reduce the computation cost and speed up the training process, we combine the transformer and CNN in our encoder, which can be seen in Fig.5. We input multi-scale features $F_3 - F_5$ to our encoder to generate more informative features. Inspired by DETR (Carion et al. (2020)), which adds a position embedding to the input feature, We first calculate the position embedding of the input features and add the position embedding to the original features $F3 - F5$ and get new features $F3^{(1)} - F5^{(1)}$. Then we input the features to a self-attention module, which can capture the local information and get $F3^{(2)} - F5^{(2)}$. After the self-attention module, we use a CNN module to increase the training process. We add the features $F3^{(1)} - F5^{(1)}$ and $F3^{(2)} - F5^{(2)}$; then we pass the result of the self-attention module to a layer normalization, then we pass the result to a $3 \times 3$ convolution layer. After the convolution, we add a group normalization and a GELU activation. Following the GELU activation, we add a $3 \times 3$ convolution layer. After the convolution layer, we restore the outputs to multi-scale features $T_3 - T_5$. Then, we flatten the $T_3 - T_5$ to a sequence and input them to the decoder. The process of the encoder can be written as

$$
\begin{aligned}
F_i^{(2)} &= \text{LN}((F_i + P_i) + \text{Att}(F_i + P_i)) \\
T_i &= \text{Conv}^3(\text{GELU}(\text{GN}(\text{Conv}^3(F_i^{(2)}))))
\end{aligned}
\tag{1}
$$

where $F_i$ is the input feature, $P_i$ is the position embedding, Att is the self-attention, LN is layer normalization, $\text{Conv}^3$ is $3 \times 3$ convolution, GELU is GELU activation, GN is group normalization.

## 4.3 DECODER

The decoder is the same as the encoder. We also combine the transformer and the convolution. We follow the same operation for the input sequence as the encoder, which first calculates the location embedding of the input features. After that, we grid the input sequence to the shape of $S \times S \times D$, then flatten them to query shapes $L \times D$ and produce a location-aware query that will provide the location information for every token. After getting the location-aware query, we input the encoder feature and location-aware query to a cross-attention layer. In the cross-attention layer, we use the location-aware query as the query and use the encoder feature as the key and value. After the cross-attention layer, we use the same normalization layer and convolution structure as the encoder to produce the decoder embedding. The process of the decoder can be written as

$$
\begin{aligned}
G &= \mathrm{LN}((Q + P_Q) + \mathrm{Cross\_Att}((Q + P_Q), (T, P_T))) \\
R &= \mathrm{Conv}^3(\mathrm{GELU}(\mathrm{GN}(\mathrm{Conv}^3(G))))
\end{aligned}
\tag{2}
$$

where $T$ is the output feature of the encoder, $P_T$ is the location embedding of the feature $T$, $Q$ is the input query, $P_Q$ is the location embedding of the input query, and $\mathrm{Cross\_Att}$ is cross attention.

## 4.4 EDGE-AWARE FEATURE FUSION MODULE (EAF)

In order to improve the accuracy of boundary prediction, we added a module called edge-aware feature fusion. This module uses the ground truth edge to combine two types of features: high-level features from the backbone network (called $F_2$) and low-level features from the encoder (called $T_3$ to $T_5$).

The edge-aware feature fusion module processes the low-level features $T_5$ to $T_3$ by first extracting information with a convolution layer, then using a binary convolutional layer to extract local binary features, which are then fed into a $1 \times 1$ convolutional layer to predict edges (called $E_5$).

Next, we up-sample the binary features to the same size as $T_4$ and concatenate them, generating a new input feature ($I_4$). We repeat this process until we reach $F_2$.

Using the edge-aware feature fusion module helps the model better recognize the boundaries of objects, leading to more precise segmentation masks and avoiding the issue of predicting one object as multiple objects. The formula for the edge-aware fusion model is given, and the output of the final block $O_2$ is sent to the mask prediction head.

$$
\begin{aligned}
O_i &= \mathrm{BCNN}(\mathrm{Conv}^{\mathrm{multi}}(I_i)) \\
E_i &= \mathrm{Conv}^1(O_i)
\end{aligned}
\qquad
I_i = \begin{cases}
T_5 & i = 5 \\
\mathrm{UP}(O_{i+1}) + T_i & i = 3, 4 \\
\mathrm{UP}(O_{I+1}) + F_i & i = 2
\end{cases}
\tag{3}
$$

where $BCNN$ is binary convolution layer, $\mathrm{Conv}^{\mathrm{multi}}$ is multi-scale convolution layer, $\mathrm{Conv}^1$ is $1 \times 1$ convolution layer, UP means up-sampling $O_i$ is the output of $i_{th}$ block and $I_i$ is the input of $i_{th}$ block and $E_i$ the the edge prediction of the $i_{th}$ block. We also output the result of the final block $O_2$ to the mask prediction head.

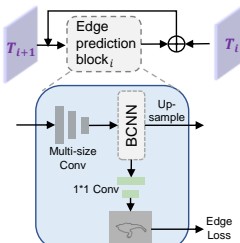

Figure 6: Our edge-aware feature fusion module uses a pyramid structure. The main component of our edge-aware feature fusion module is an edge prediction block. Given the input feature, we use a multi-size convolution following a BCNN layer to extract its feature. Then, we up-sample the result to the same size as the next input feature size. We use a 1 convolution layer to predict the edge and use the ground truth edge as supervision.

## 4.5 MASK PREDICTION HEAD

We use the same structure as OSFormer (Pei et al. (2022)). More detail can be seen in the supplement.

## 4.6 Loss function

Our loss function contains three parts, edge loss, location loss, and mask loss. We use dice loss for the edge loss and location loss; for mask loss, we use focal loss. Hence, our final loss function can be written as

$$L = \lambda_{edge}L_{edge} + \lambda_{location}L_{location} + \lambda_{mask}L_{mask}$$

In our experiments, $\lambda_{edge}$ and $\lambda_{location}$ is set to 1 while $\lambda_{mask}$ is set to 3 to balance different loss.

## 5 Experiments

### 5.1 Experimental setup

**datasets** Our experiments use two benchmark datasets: COD10K (Fan et al. (2020)) and NC4K (Lv et al. (2021)). The COD10K datasets include 3040 training images with instance-level annotations and 2026 for testing. The NC4K datasets contain 4121 images with instance-level labels. We use the COD10K training set to train our model and test our dataset on the COD10K testing set and NC4K dataset. In order to provide more training samples for the model, we resize the input images to multiple sizes. We guarantee that the size of the shorter side is between 480 and 800, and the longer side of the input image is less than 1333 after resizing.

**evaluation metrics** Our experiments use COCO-style evaluation metrics, including $AP$, $AP_{50}$ and $AP_{75}$, but our experiments have slight differences. The original COCO evaluation metrics use mAP, which will calculate the mean AP for every category. However, our camouflaged datasets are class-agnostic. Hence, we only need to calculate the AP for the whole dataset while ignoring the category.

**implement details** We implemented our $\alpha$-Former using PyTorch and trained it on a single V100-sxm2. To build our model, we utilized ResNet-50 (He et al. (2016)) as the backbone, which had been pre-trained with the ImageNet (Deng et al. (2009)) dataset. We trained our model for 90K iterations with a batch size of 2 during our experiments. The optimizer we used was SGD, with an initial learning rate of $2.5e-4$, and we reduced the learning rate by a factor of $0.1$ when the number of iterations reached 60K and 80K. The weight decay parameter was set to $1e-4$.

### 5.2 Comparison with the State-of-the-arts

We compare our model with current State-of-the-art models. Because there are not many camouflaged instance segmentation models, we also use several generic instance segmentation models and limit these models to train and test on the camouflaged datasets. In order to have fair comparisons, we use pre-trained ResNet-50 as the backbone for all models. The results are shown in Table.1

Table 1: Quantitative results of the $\alpha$-Former, the best results are highlighted in **bold**.

| method | COD10K | | | NC4K | | |
|---|---|---|---|---|---|---|
| | AP | AP50 | AP75 | AP | AP50 | AP75 |
| Mask-RCNN (He et al. (2017)) | 25.0 | 55.5 | 20.4 | 27.7 | 58.6 | 22.7 |
| MS-RCNN (Huang et al. (2019)) | 30.1 | 57.5 | 25.7 | 36.1 | 68.9 | 33.5 |
| Cascade RCNN (Cai & Vasconcelos (2019)) | 25.3 | 56.1 | 21.3 | 29.5 | 60.8 | 24.8 |
| HTC (Chen et al. (2019)) | 28.1 | 56.3 | 25.1 | 29.8 | 59.0 | 26.6 |
| Mask Transfiner (Ke et al. (2022)) | 28.7 | 56.3 | 26.4 | 29.4 | 56.7 | 27.2 |
| YOLACT (Bolya et al. (2019)) | 24.3 | 53.3 | 19.7 | 32.1 | 65.3 | 27.9 |
| CondInst (Tian et al. (2020)) | 30.6 | 63.6 | 26.1 | 33.4 | 67.4 | 29.4 |
| QueryInst (Fang et al. (2021)) | 28.5 | 60.1 | 23.1 | 33.0 | 66.7 | 29.4 |
| SOTR (Guo et al. (2021)) | 27.9 | 58.7 | 24.1 | 29.3 | 61.0 | 25.6 |
| SOLOv2 (Wang et al. (2020)) | 32.5 | 63.2 | 29.9 | 34.4 | 65.9 | 31.9 |
| OSFormer (Pei et al. (2022)) | 41.0 | 71.1 | 40.8 | 42.5 | 72.5 | 42.3 |
| $\alpha$-Former(Ours) | **42.5** | **72.8** | **41.8** | **42.9** | **72.9** | **43.3** |

### 5.3 Ablation Study

#### 5.3.1 Adapter

In this section, we show the improvement of adding the feature aggregation adapter to our feature extractor. The target for our adapter is to provide the extra local feature to our encoder. If we directly delete the adapter, the input domain will be different, and the pre-trained backbone cannot deal with the input with the local feature. However, to provide a fair comparison, we still need to provide the local feature to the encoder-decoder and the edge-aware fusion module. Hence, we concatenate our local features to the ResNet extracted features and change the input channel numbers of the encoder

and edge-aware fusion module. In this way, we can still provide the local features to the encoder and edge-aware fusion module and provide a fair comparison. Also, we tried a different setting that modified the first layer of the pre-trained backbone and randomly initialized (RI) this layer to demonstrate the efficiency of our adapter. To better demonstrate the efficiency of our adapter, We also test the adapter on the traditional descriptor. The results are shown in Table.2. The results show that our adapter is helpful for the binary filter and can improve the performance of the traditional descriptor.

Table 2: Ablations for the existence of feature aggregation adapter.

| method | COD10K | | | NC4K | | |
|---|---|---|---|---|---|---|
| | AP | AP50 | AP75 | AP | AP50 | AP75 |
| HOG + RI | 36.785 | 63.585 | 37.906 | 35.474 | 64.150 | 37.246 |
| HOG w/o adapter | 40.801 | 70.435 | 41.407 | 42.682 | 72.647 | 43.154 |
| HOG w/ adapter | 40.934 | 70.887 | 40.285 | 42.765 | 71.988 | 44.226 |
| LBP + RI | 33.562 | 61.623 | 34.732 | 35.631 | 64.463 | 35.462 |
| LBP w/o adapter | 39.530 | 69.419 | 39.331 | 42.288 | 71.077 | 42.162 |
| LBP w/ adapter | 40.410 | 70.323 | 40.184 | 41.794 | 71.313 | 42.484 |
| Circle-LBP + RI | 35.246 | 66.352 | 36.462 | 36.853 | 67.432 | 35.241 |
| Circle-LBP w/o adapter | 40.270 | 70.550 | 40.257 | 42.668 | 73.669 | 42.172 |
| Circle-LBP w/ adapter | 40.424 | 69.622 | 40.764 | 41.921 | 71.661 | 42.133 |
| Binary filter + RI | 36.415 | 64.151 | 35.414 | 33.541 | 67.252 | 34.532 |
| Binary filter w/o adapter | 41.427 | 71.247 | 40.984 | 42.610 | 71.517 | 42.985 |
| Binary filter w/ adapter | 42.453 | 72.735 | 41.758 | 42.936 | 72.905 | 43.278 |

Table 3: Ablations for the existence of edge-aware feature fusion module.

| method | COD10K | | | NC4K | | |
|---|---|---|---|---|---|---|
| | AP | AP50 | AP75 | AP | AP50 | AP75 |
| HOG w/o EAF | 37.658 | 66.584 | 35.984 | 39.252 | 67.971 | 38.756 |
| HOG w/ EAF | 40.934 | 70.887 | 40.285 | 42.765 | 71.988 | 44.226 |
| LBP w/o EAF | 36.128 | 67.197 | 36.725 | 36.375 | 68.258 | 37.813 |
| LBP w/ EAF | 40.410 | 70.323 | 40.184 | 41.794 | 71.313 | 42.484 |
| Circle-LBP w/o EAF | 35.254 | 64.741 | 36.194 | 36.581 | 66.943 | 36.135 |
| Circle-LBP w/ EAF | 40.424 | 69.622 | 40.764 | 41.921 | 71.661 | 42.133 |
| Binary filter w/o EAF | 38.019 | 69.765 | 36.813 | 37.083 | 68.672 | 38.731 |
| Binary filter w/ EAF | 42.453 | 72.735 | 41.758 | 42.936 | 72.905 | 43.278 |

### 5.3.2 EDGE-AWARE FEATURE FUSION MODULE

This section provides the ablation study of our edge-aware fusion module. Our edge-aware fusion module can provide precise boundary prediction information to the final prediction heads. Similar to the adapter, we show the results using different descriptors, including traditional descriptors and our binary filter. The results are shown in Table.3. The results show that our proposed edge-aware feature fusion module can improve the performance for about 4 AP higher than the model that does not have an edge-aware feature fusion module. It shows the efficiency of our edge-aware feature fusion module and proves that edge prediction is crucial in camouflaged instance segmentation. The qualitative results of our edge-aware feature fusion module can be seen in Fig.7, which shows that our edge-aware feature fusion module can deal with different scenarios and precisely predict the edge of the target object.

### 5.3.3 COMPARISON WITH THE TRADITIONAL DESCRIPTOR

As shown in Table.4, we compare the performance of our binary filter and the traditional descriptor. Here, Baseline means no descriptors are added. Because SIFT cannot generate a feature map that has the same size as the original images, in order to use the same architecture and have a fair comparison, we mainly focus on the HOG (Dalal & Triggs (2005)), LBP (Ojala et al. (1994)), circle-LBP (Ojala et al. (2002)) descriptor in our experiments. Except for the local feature extractor, our experiments' other settings are the same. We can see that some of the traditional descriptors can outperform the model that does not include any local feature extractor. However, our learnable binary filter can perform better than the traditional descriptor. This experiment demonstrates our binary filter's efficiency and ability to provide powerful local features to improve the performance of the model.

Table 4: Comparison with the traditional descriptor, the best results are highlighted in **bold**.

| method | COD10K | | | NC4K | | |
|---|---|---|---|---|---|---|
| | AP | AP50 | AP75 | AP | AP50 | AP75 |
| Baseline | 40.244 | 69.875 | 39.422 | 41.718 | 71.640 | 41.179 |
| HOG | 40.934 | 70.887 | 40.285 | 42.765 | 71.988 | **44.226** |
| LBP | 40.410 | 70.323 | 40.184 | 41.794 | 71.313 | 42.484 |
| Circle-LBP | 40.424 | 69.622 | 40.764 | 41.921 | 71.661 | 42.133 |
| Binary filter | **42.453** | **72.735** | **41.758** | **42.936** | **72.905** | 43.278 |

Table 5: Performance of $\alpha$-Former with different kernel size in the binary filter, the best results are highlighted in **bold**.

| method | COD10K | | | NC4K | | |
|---|---|---|---|---|---|---|
| | AP | AP50 | AP75 | AP | AP50 | AP75 |
| $3 \times 3$ | **42.453** | **72.735** | **41.758** | **42.936** | **72.905** | **43.278** |
| $5 \times 5$ | 41.308 | 70.624 | 41.707 | 42.567 | 72.075 | 43.198 |
| $7 \times 7$ | 40.476 | 70.047 | 40.790 | 42.136 | 71.895 | 42.698 |
| $9 \times 9$ | 40.691 | 70.116 | 40.810 | 41.164 | 71.043 | 42.580 |

### 5.3.4 INFLUENCE OF DIFFERENT KERNEL SIZE IN BCNN

This section explores the influence of different kernel sizes in our binary filter. Different kernel sizes will have different receptive fields, and a larger receptive field will provide more pixels in

one convolution operation. In our binary filter, it will affect the final local binary feature of the binary filter. Our results are shown in Table.5. It shows that a smaller kernel size can have better performance. The reason that small kernel sizes have better performance may be that camouflaged objects have similar pixel values as the background. The larger kernel may increase the influence of the background and result in final performance drops.

## 5.4 VISUALIZATIONS

This section presents the qualitative results of the $\alpha$-Former, including the edge prediction achieved by our edge-aware fusion module. The results demonstrate the effectiveness of our approach, as our module can predict precise boundaries, as shown in the second row's first column, where it accurately identifies the feet of a challenging target object. Additionally, our $\alpha$-Former can successfully handle different backgrounds, such as branches, land, or aquatic plants, and precisely segment different target objects, including birds, fishes, and terrestrial animals. Moreover, our model can generate accurate edges even when the target object is partially occluded, as seen in the last row's first column. This suggests that our approach can extract semantic information from the backbone's features and recognize the object as the same entity, even if it is not continuous. Overall, these results demonstrate the robustness and effectiveness of our $\alpha$-Former in challenging scenarios.

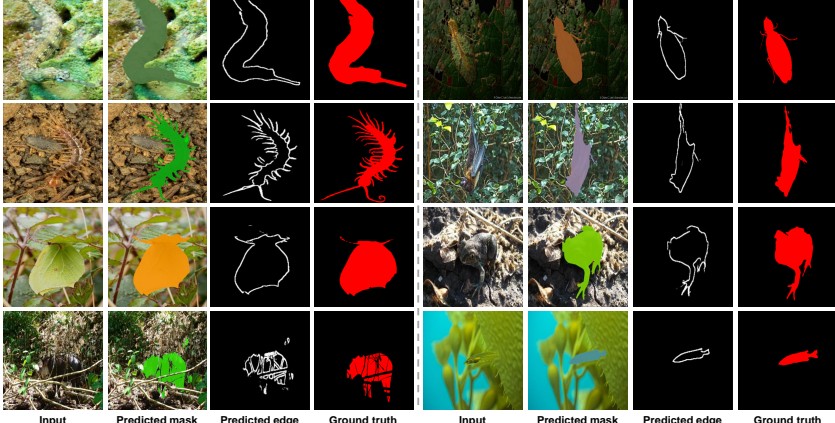

Figure 7: The results of our $\alpha$-Former's qualitative evaluation demonstrate its ability to extract precise boundaries and its strong performance in a range of challenging scenarios. These findings suggest that our proposed approach can effectively address the complexities of real-world image segmentation tasks.

## 6 CONCLUSION

In conclusion, we contribute a novel local feature-aware transformer framework called $\alpha$-Former targeting on camouflaged instance segmentation. Observing the camouflaged objects' characteristics, we find that traditional descriptors are sensitive to the camouflaged objects. Inspired by the traditional descriptor, we design a novel binary filter to extract the camouflaged image's local features. To provide the local features to the encoder, we design a feature aggregation adapter to fuse the pre-trained backbone and the local features input. Besides, we design an edge-aware feature fusion module to improve the boundary prediction of the camouflaged object by combining multi-level features and utilizing the ground truth edge as the supervision. Moreover, we design multiple ablation studies to show the effectiveness of our proposed binary filter, feature aggregation adapter, and edge-aware feature fusion module. We also provide the qualitative results of our $\alpha$-Former to show our robustness to different backgrounds. Moreover, $\alpha$-Former is very easy to train; we only need about 3000 training images, and it takes about one day to finish the training process. We believe the $\alpha$-Former is a new state-of-the-art for camouflaged instance segmentation, and it can be transferred to applications like medical diagnosis, photo-realistic blending, etc.

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
