# Supplement of $\alpha$-Former:Local-Feature-Aware(L-FA) Transformer

## 1 More Implement Details

### 1.1 More Details of the Feature Aggregation Adapter

Our feature aggregation adapter uses a tiny initial value to guarantee at the beginning of the training, the output domain is the same as the input image domain. Specifically, we set the mean and the variance value of the convolution weight as 0 and 0.001, and the bias value of the convolution layer as 0. Using the tiny-initialized convolution layer and the skip connection, we can know that the output of the adapter is almost the same as the input at the beginning of the training.

### 1.2 More Details of the Edge-aware Feature Fusion Module

In this section, we provide more details about our edge-aware feature fusion module. Our edge-aware feature fusion module uses multi-scale features to predict the boundary of the target object. As shown in table.1, we provide the input and output shapes of the different edge prediction blocks.

Table 1: Input and output shape of different edge prediction block

| Block | Input Shape | Output Shape |
|-------|-------------|--------------|
| $\text{block}_5$ | $\frac{H \times W}{32}$ | $\frac{H \times W}{16}$ |
| $\text{block}_4$ | $\frac{H \times W}{16}$ | $\frac{H \times W}{8}$ |
| $\text{block}_3$ | $\frac{H \times W}{8}$ | $\frac{H \times W}{4}$ |
| $\text{block}_2$ | $\frac{H \times W}{4}$ | $\frac{H \times W}{4}$ |

### 1.3 More Details of the Prediction Head

In this section, we provide more details about our prediction head. We follow the same architecture as OSFormerPei et al. (2022). As shown in Fig.1. During the training process, we use a fully-connected layer to calculate the location label. At the same time, we use a multi-layer perceptron to calculate the instance-aware parameters. Then we assign positive and negative locations using ground truth. During the testing process, we use a confidence score of the location label to filter ineffective parameters of the instance-aware parameters. Then we use two linear layers to calculate the weight and bias to calculate the segmentation mask. Then we use an up-sampling operation to get the final prediction masks.

## 2 More Visualizations

As shown in Fig.2, we provide more visualizations in this section.

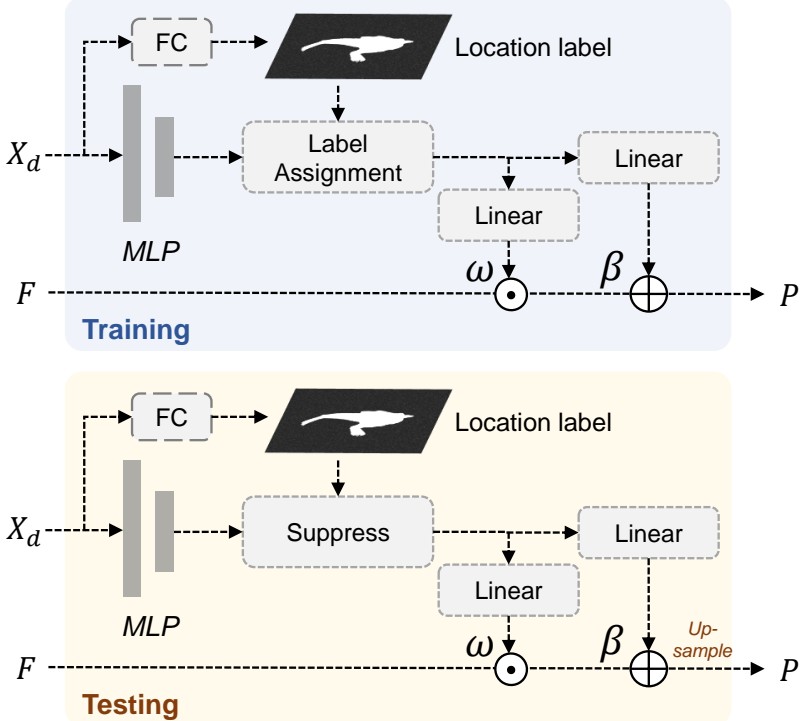

Figure 1: During the training process, our prediction head uses location labels as supervision, and during the testing process, our prediction head uses location labels to filter ineffective parameters.

## REFERENCES

Jialun Pei, Tianyang Cheng, Deng-Ping Fan, He Tang, Chuanbo Chen, and Luc Van Gool. Osformer: One-stage camouflaged instance segmentation with transformers. In *European Conference on Computer Vision*, pp. 19–37. Springer, 2022.

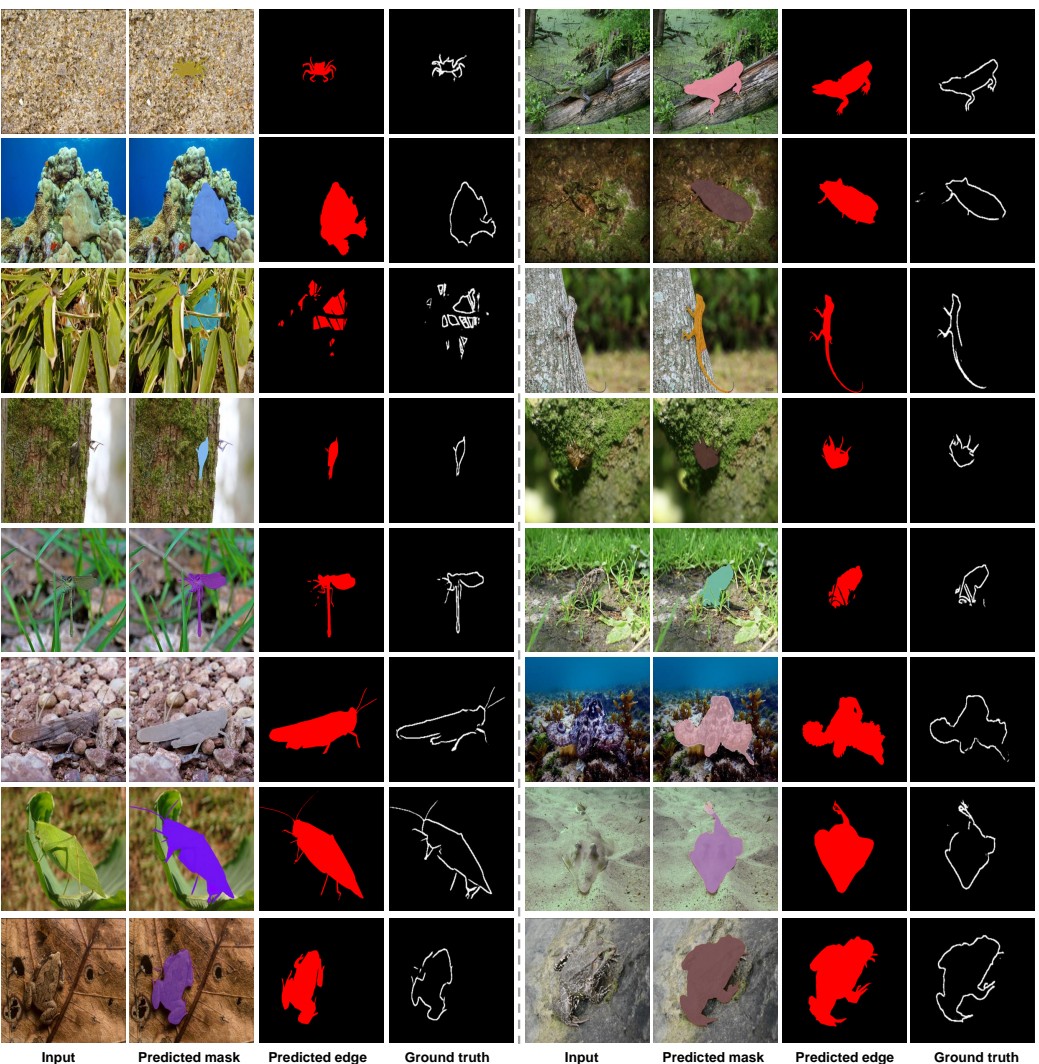

Figure 2: The qualitative results of $\alpha$-Former.