# OpenReview forum: "α-Former: Local-Feature-Aware (L-FA) Transformer"
_ICLR.cc/2024/Conference — ICLR 2024 Conference Withdrawn Submission_

### Official Review · Reviewer_1G3o · 2023-10-29

**Soundness:** 2 fair
**Presentation:** 2 fair
**Contribution:** 2 fair
**Rating:** 3
**Confidence:** 5

**Summary:**

This paper proposes a novel method to solve the camouflaged instance segmentation (CIS) task. By considering the importance of object boundary information and inspired by traditional computer vision descriptors, this paper designs a novel binary filter to extract local information, a feature aggregation adapter to fuse local information and original image information, and an edge-aware feature fusion module to improve the boundary prediction. Combining these designs, this paper achieves state-of-the-art performance on the camouflaged instance segmentation (CIS) task.

**Strengths:**

* State-of-the-art performance on the camouflaged instance segmentation (CIS) task.
* Reasonable module design, such as learnable binary filter.

**Weaknesses:**

* The effectiveness of the feature aggregation adapter is limited.
According to Table 2 in this paper, using adapter has a negative impact on the performance of traditional descriptors on NC4K. For example, AP of (LBP w/o adapter) 42.288, AP of LBP w/ adapter 41.794. And same situtation for Circle-LBP. Even for the Binary filter, the improvement is very limited. (42.610->42.936).
* Typo: There exists a mismatch figure caption between Figure 7 in this paper and Figure 2 in supplementary_material. (Predicted edge and Ground truth).
* Complex pipeline and missing ablation study: This paper introduces attention module, positional encoding, encoder, decoder, but missing results to demenstrate the effectiveness of each components and how each component help the camouflaged instance segmentation (CIS) task.

**Questions:**

See Weakness.

---

### Official Review · Reviewer_vJX5 · 2023-10-31

**Soundness:** 2 fair
**Presentation:** 3 good
**Contribution:** 2 fair
**Rating:** 3
**Confidence:** 4

**Summary:**

This paper proposes a Vision Transformer based deep learning model, dubbed as $\alpha$-FORMER, for the task of camouflaged instance segmentation (CIS).
Its key idea is to leverage hand-crafted local features into the modern deep learning modules, and feed the fused features into the CIS task head.
Extensive experiments on two standard datasets, COD10K and NC4K, show its state-of-the-art performance against prior art.
Some ablation studies and further analysis are also conducted.

**Strengths:**

- This paper is well-written, easy-to-follow and well-organized.
- The performance shows a clear improvement of the compared prior arts under all the experimental settings.

**Weaknesses:**

- The technique contribution of this work, to either the field of CIS or COD, from the reviewer's view, is incremental, and in many aspects,
exaggerated. Reasons:
(1) The claim of 'some traditional descriptors are sensitive to the boundary of camouflaged objects' is of course not firstly noticed by the authors. For every benchmark in COD or even older SOD, the traditional hand-crafted methods show clear performance decline on these tasks. Thus, there is no surprise that, on a variation task CIS from COD and SOD, such methods show clear performance drop.
(2) The methodology and module design on feature fusion or boundary enhancement has little to do with the instances. In other words, they can be directly used for sister tasks such as COD or mother tasks such as SOD.

- The module design is very ordinary, and lack of theoretical insights. For example, learnable local binary filter has been studied six years ago. The below key reference and many subsequent works have already used this design.

[1] Juefei-Xu, Felix, Vishnu Naresh Boddeti, and Marios Savvides. "Local binary convolutional neural networks." Proceedings of the IEEE conference on computer vision and pattern recognition. 2017.

- The idea to jointly use and fuse hand-crafted features and deep learning features has also been extensively studied in many vision tasks. So, the claimed key innovation of the proposed method is also incremental. For example:

[2] Song, Tiecheng, et al. "Quaternionic extended local binary pattern with adaptive structural pyramid pooling for color image representation." Pattern Recognition 115 (2021): 107891.

[3] Hubálovský, Štěpán, et al. "Evaluation of deepfake detection using YOLO with local binary pattern histogram." PeerJ Computer Science 8 (2022): e1086.

- CIS is a task incremental to the prior COD, or the even older task SOD. Compared with COD methods, the only difference of model design can be the task-specific head and the evaluation metric, just as semantic segmentation and instance segmentation. So, (1) Table 1 should compare more recent COD methods by shifting them into the CIS task head. (2) Besides, the reviewer argues it is necessary, as the author has already compared some segmentation models such as MaskFormer, Mask-RCNN and etc. (3) Then, how about the performance of some more recent segmentation SOTA, like Mask2Former and OneFormer?

**Questions:**

The reviewer would insist clear reject this paper after rebuttal, unless there are some very strong arguements to address the weakness part, such as:

- Overclaimed contribution. For example, 'first notice some traditional descriptors are sensitive to the boundary of camouflaged objects'. For every benchmark in COD or even older SOD, the traditional hand-crafted methods show clear performance decline on these tasks. Thus, there is no surprise that, on a variation task CIS from COD and SOD, such methods show clear performance drop.

- The methodology and module design on feature fusion or boundary enhancement has little to do with the instances. In other words, they can be directly used for sister tasks such as COD or mother tasks such as SOD. Little insights on CIS itself can be leveraged from this work.

- The module design is very ordinary, and lack of theoretical insights. For example, learnable local binary filter has been studied six years ago. The below key reference and many subsequent works have already used this design.

[1] Juefei-Xu, Felix, Vishnu Naresh Boddeti, and Marios Savvides. "Local binary convolutional neural networks." Proceedings of the IEEE conference on computer vision and pattern recognition. 2017.

- The idea to jointly use and fuse hand-crafted features and deep learning features has also been extensively studied in many vision tasks. So, the claimed key innovation of the proposed method is also incremental. For example:

[2] Song, Tiecheng, et al. "Quaternionic extended local binary pattern with adaptive structural pyramid pooling for color image representation." Pattern Recognition 115 (2021): 107891.

[3] Hubálovský, Štěpán, et al. "Evaluation of deepfake detection using YOLO with local binary pattern histogram." PeerJ Computer Science 8 (2022): e1086.

- CIS is a task incremental to the prior COD, or the even older task SOD. Table 1 should compare more recent COD methods by shifting them into the CIS task head.

- As the author has already compared some segmentation models such as MaskFormer, Mask-RCNN and etc. Then, how about the performance of some more recent segmentation SOTA, like Mask2Former and OneFormer? Is the proposed model still able to outperform them?

---

### Official Review · Reviewer_Zqb6 · 2023-11-01

**Soundness:** 2 fair
**Presentation:** 2 fair
**Contribution:** 2 fair
**Rating:** 3
**Confidence:** 4

**Summary:**

The work proposed \alpha-Former that incorporates a fixed-weight convolution layer (BCNN) to extract edge features in camouflaged instances.

The proposed network contains 5 components: (1) A feature extractor with a binary filter to extract similar features as the LBP; (2) A transformer encoder (3) An edge-aware feature fusion module to generate precise boundaries. (4) A transformer decoder to extract the information from the embedding (5) Mask predict head to predict final instance mask.

To evaluate the method's performance, it was benchmarked on the COD10K and NC4K datasets using metrics such as AP, AP50, and AP75.

**Strengths:**

Incorporate edge detector to reserve edge information.

**Weaknesses:**

The paper presents several limitations, which are outlined below:

Effectiveness of Proposed Method: The method combines a traditional pre-defined feature extractor with CNN and Transformer components. However, it lacks a convincing demonstration of the effectiveness of this approach.

Lack of Related Work Discussion: There is a notable absence of discussion regarding related work. The incorporation of edge-aware networks into segmentation networks to address camouflaged instance segmentation has been a popular research direction. Notable works, such as those by [1,2,7], have explored this area.

Limited Benchmarking Datasets: The method was benchmarked on only two datasets, COD10K and NC4K. To provide a comprehensive evaluation, common datasets for benchmarking camouflaged object detection, such as CAMO and CHAMELEON, should also be included.

Inappropriate Evaluation Metrics: The paper utilizes Average Precision (AP) as the evaluation metric for camouflaged instance segmentation. However, commonly accepted metrics for this task include structure-measure, mean E-measure, weighted F-measure, and mean absolute error.

Lack of Recent Comparative Analysis: The paper does not include a comparison with recent works on Camouflaged Object Detection, such as the works by [4-9]. This omission limits the paper's ability to demonstrate its novelty and competitiveness in the field.

In addressing these limitations, the paper could strengthen its contribution and provide a more comprehensive assessment of its proposed method.
[1] Sun, Yujia, Shuo Wang, Chenglizhao Chen, and Tian-Zhu Xiang. "Boundary-guided camouflaged object detection." IJCAI 2022.
[2] Sun, Dongyue, Shiyao Jiang, and Lin Qi. "Edge-Aware Mirror Network for Camouflaged Object Detection." ICME 2023.
[3] Jiang, Shiyao, Xinyue Li, Miao Yang, and Lin Qi. "Edge-Aware Fusion for Camouflaged Object Detection." ICIVC 2022.
[4] Pang, Youwei, Xiaoqi Zhao, Tian-Zhu Xiang, Lihe Zhang, and Huchuan Lu. "Zoom in and out: A mixed-scale triplet network for camouflaged object detection." CVPR 2022.
[5] Zhong, Yijie, Bo Li, Lv Tang, Senyun Kuang, Shuang Wu, and Shouhong Ding. "Detecting camouflaged object in frequency domain." CVPR 2022.
[6] Jia, Qi, Shuilian Yao, Yu Liu, Xin Fan, Risheng Liu, and Zhongxuan Luo. "Segment, magnify and reiterate: Detecting camouflaged objects the hard way." CVPR 2022.
[7] He, Chunming, Kai Li, Yachao Zhang, Longxiang Tang, Yulun Zhang, Zhenhua Guo, and Xiu Li. "Camouflaged object detection with feature decomposition and edge reconstruction, CVPR2023
[8]Huang, Zhou, Hang Dai, Tian-Zhu Xiang, Shuo Wang, Huai-Xin Chen, Jie Qin, and Huan Xiong. "Feature shrinkage pyramid for camouflaged object detection with transformers." CVPR2023
[9] Hu, Xiaobin, Shuo Wang, Xuebin Qin, Hang Dai, Wenqi Ren, Donghao Luo, Ying Tai, and Ling Shao. "High-resolution iterative feedback network for camouflaged object detection, AAAI 2023

**Questions:**

See the limitations